# Effect of Ethnicity on Changes in Fat and Carbohydrate Oxidation in Response to Short-Term High Intensity Interval Training (HIIT): A Pilot Study

**DOI:** 10.3390/ijerph18084314

**Published:** 2021-04-19

**Authors:** Todd A. Astorino, Jamie L. De Revere

**Affiliations:** Department of Kinesiology, California State University San Marcos, San Marcos, CA 92096-0001, USA; jldrev@gmail.com

**Keywords:** interval training, fat metabolism, blood lactate concentration, ethnicity, minimum difference, individual responsiveness to training

## Abstract

This study compared changes in substrate metabolism with high intensity interval training (HIIT) in women of different ethnicities. Twelve Caucasian (C) and ten Hispanic women (H) (age = 24 ± 5 yr) who were inactive completed nine sessions of HIIT at 85 percent peak power output (%PPO). Pre-training, changes in fat oxidation (FOx) and carbohydrate oxidation (CHOOx) during progressive cycling were measured on two days to compute the minimum difference (MD). This test was repeated after the last training session. Between baseline tests, estimates of FOx and CHOOx were not different (*p* > 0.05) and were highly related (intraclass correlation coefficient equal to 0.72 to 0.88), although the coefficient of variation of maximal fat oxidation (MFO) was equal to 30%. Training significantly increased MFO (*p* = 0.03) in C (0.19 ± 0.06 g/min to 0.21 ± 0.06 g/min, *d* = 0.66) and H (0.16 ± 0.03 g/min to 0.19 ± 0.03 g/min, *d* = 1.3) that was similar (*p* = 0.92) between groups. There was a significant interaction for FOx (*p* = 0.003) as it was only increased in H versus C, although both groups exhibited reduced CHO oxidation (*p* = 0.002) with training. Use of MD revealed that only 3 of 22 women show meaningful increases in MFO (>0.08 g/min). The preliminary data reveals that a small dose of low-volume HIIT does not alter fat and CHO oxidation and there is little effect of ethnicity on the response to training.

## 1. Introduction

Completion of moderate intensity continuous training (MICT) improves cardiorespiratory fitness (VO_2_max) and health status in adults [1], yet attainment of 150 min/wk of MICT or 75 min/wk of vigorous exercise is low in adults living in the United States (25%, CDC 2020). One barrier to regular physical activity is a lack of time [2], as the current Physical Activity guidelines [1] recommend 30 min/d of MICT on most days of the week. Recently, high intensity interval training (HIIT) has been widely investigated for its health and fitness-promoting benefits as a potential alternative to MICT. Data reveal that HIIT elicits similar [3] and in some cases superior adaptations versus MICT [4]. One advantage of HIIT versus MICT is its higher intensity, eliciting 85–100 percent of maximal heart rate (%HRmax) [5] that enables the exerciser to accumulate a higher volume of near-maximal exercise versus MICT. It is plausible that the higher intensity characteristic of HIIT elicits more rapid glycogen degradation leading to enhanced activation of peroxisome–proliferator activated receptor ¥ coactivator (PGC-1α) [6], which is viewed as the primary mediator of skeletal muscle mitochondrial biogenesis [7]. In addition, results from Little et al. [8] reveal that acute bouts of HIIT increase the nuclear abundance of PGC-1α that is consequent with greater mRNA expression of various mitochondrial genes.

Despite the efficacy of HIIT, Astorino and Schubert [9] and Gurd et al. [10] identified individual responses to HIIT and its more intense form, sprint interval training, as some participants reveal marked increases in various health-related outcomes; whereas, others reveal little to no response to the identical regimen. This is concerning as regular physical activity is used to combat severity of chronic disease or reduce its incidence. In 481 sedentary adults, approximately 50% of the heterogeneity in the VO_2_max response to endurance training is explained by genetics [11], yet the other 50% is likely related to participants’ sleep and dietary habits, habitual physical activity, and traits of exercise training [12]. However, scientists still do not understand why this individual response to exercise training occurs, so further research is needed.

The HERITAGE study examined genomic responses to 20 wk of MICT in Caucasian (C) and African American (AA) adults [11]. Data showed significant increases in VO_2_max and reductions in blood pressure, although individual responses were apparent. Subsequent large-scale training studies, including STRRIDE [13], examining the effect of MICT on various health-related markers rarely included populations other than C or AA, so it is unknown whether adults of other ethnicities respond similarly to physical activity as C and AA. Recently, Gill et al. [14] stated that the minimum level of physical activity needed to confer health benefits across persons of various ethnicities may be unequal. Moreover, they stated that innate differences in VO_2_max and capacity for fat oxidation potentially contribute to ethnic differences in the cardiometabolic risk profile, and that ethnicity-specific physical activity guidelines are warranted. Hispanics (H) exhibit lower physical activity compared to C [15], which may impact their health status and augment potential risk of chronic disease. Overall, further work is needed to examine if physical activity also benefits health in H adults.

Moderate intensity continuous training [16] and HIIT [17] increase fat oxidation (FOx) assessed during exercise. Increases in oxidative capacity represented by enhanced activities of citrate synthase [18,19,20], β-HAD [18,19], and cytochrome c-oxidase [19] are frequently reported in response to interval training, suggesting the potential for enhanced capacity for fat oxidation. This adaptation is important considering the relationship between FOx and weight gain [21], as well as insulin sensitivity and maximal fat oxidation (MFO) [22], which are associated with metabolic health [23]. However, the increase in FOx observed in response to HIIT is not as prevalent as the widely documented increase in VO_2_max [24], which may be due to substantial day-to-day variability in the measure. In active men, Croci et al. [25] reported a coefficient of variation in MFO up to 26% in response to repeated sessions of progressive cycling. However, participants replicated their food intake and abstained from physical activity for only 24 h before the two sessions, which may be inadequate as energy metabolism is altered by differences in dietary intake and physical activity for up to 48 h before testing [26]. Similarly, Chrzanowki–Smith et al. [27] exhibited a coefficient of variation in peak fat oxidation equal to 21% in 99 adults with divergent body composition (8%–40%BF) and VO_2_max (22–66 mL/kg/min). Across these studies, marked variability in MFO occurred despite mean values being almost identical between tests. In addition, use of accelerometry preceding assessment of FOx would confirm if physical activity was similar before each session, which to our knowledge, has yet to be done in a training study.

Cortright et al. [28] reported significant, yet similar increases in FOx in C and AA women who completed ten sessions of MICT, although another study showed lower FOx in lean AA women versus lean C women performing an aerobic exercise [29]. This suggests that ethnicity may elicit discrepant metabolic responses to exercise. To our knowledge, there are little data examining changes in fat and CHO oxidation in H adults, and no study has identified changes in FOx in response to HIIT, as frequently reported in C. The present study compared changes in fat and carbohydrate oxidation in response to low-volume HIIT in H and C women who are more prone to obesity than men. We advanced this topic by examining day-to-day variability in fat and CHO oxidation in inactive adults through implementing repeated baseline tests [30], and utilized these data to identify individuals revealing meaningful increases in FOx in response to training. We hypothesized that C and H women would reveal significant increases in FOx in response to training, which was similar to Cortright et al. [28].

## 2. Materials and Methods

Design: Prior to training, participants underwent two baseline tests of VO_2_max and fat and CHO oxidation across four separate days, with these sessions separated by a minimum of 48 h and up to 96 h. Sessions were held at the same time of day within participants and required women to be well-rested, hydrated, fasted for >3 (VO_2_max testing) and 6 h (assessment of fat and CHO oxidation), and abstain from physical activity for 48 h prior, which were confirmed with a survey. Subsequently, women performed nine sessions of low-volume HIIT that were held over a three week period. At least 48 h after the final training session, baseline tests were repeated following identical procedures. Figure 1 shows our experimental design. Habitual physical activity was monitored at baseline, during training, and prior to post-testing.

Participants: Twenty eight healthy, inactive (<150 min/week of physical activity in the last year), non-obese (BMI < 30 kg/m^2^) women ranging from 19–35 years old were recruited. Six women (four C and two H) withdrew after completing initial testing, so 22 women completed the study. The women were either 100% Caucasian (n = 12) or 100% Hispanic (n = 10), which was identified by self-report. We recorded the ethnicity of their parents and grandparents, and all had to have the same ethnicity for women to be included. Participants did not smoke, were not taking any medications, and had no physical condition that modified their training response. Prior to baseline testing, women completed a health history and physical activity questionnaire (IPAQ) to confirm that they met these guidelines. Participants provided written informed consent before participating in this study, whose protocol was approved by the University Institutional Review Board. These data originated from a larger study examining the cardiorespiratory and hemodynamic adaptation to HIIT in inactive women [31].

Assessment of VO_2_max: Height and body mass were assessed using a balance scale and stadiometer. Subsequently, women initiated VO_2_max testing on an electronically-braked cycle ergometer (Velotron DynaFit Pro, Quark, Spearfish, SD, USA) using a ramp protocol. Individual adjustments on the cycle ergometer including seat, bar, and handlebar height were adjusted for each participant during this session and maintained for subsequent trials. Power output began at 30 or 40 W for 2 min followed by a 15 or 20 W/min increase in power output until volitional exhaustion. During exercise, subjects expired through a plastic mouthpiece and low resistance three-way valve into tubing connected to a mixing chamber. Measures of ventilation and expired fractions of oxygen and carbon dioxide were obtained throughout exercise by a metabolic cart (ParvoMedics True One, Sandy, UT, USA). Volume of oxygen uptake (VO_2_), carbon dioxide production (VCO_2_), and ventilation (V_E_) were time-averaged every 15 s. The test was terminated when the pedal cadence was below 50 rev/min. Peak power output (PPO) was identified as the value coincident with exhaustion and was used to set intensities for HIIT, which was the average of the PPO values from both trials. Subsequently, participants recovered for 10 min before undergoing a verification test to confirm VO_2_max attainment. After 2 min of cycling at 10%PPO, participants pedaled “all-out” at 105%PPO as this constant load test has been identified as a robust approach to verify VO_2_max attainment [32]. A ‘true’ VO_2_max was attained in all participants at all time points of testing, as the verification value was less than 3% higher than the ramp value [33]. Baseline VO_2_max was identified as the average of all four values acquired, and post-training data were represented by the average of the ramp and verification value.

Assessment of fat and CHO oxidation: After 5 min of seated rest to allow respiratory exchange ratio (RER) to stabilize, women cycled for 6 min at 10%PPO followed by successive 5 min stages at 20, 30, 40, and 50%PPO. Participants breathed into a mask covering the nose and mouth (Hans Rudolph, Kansas City, MO, USA), which was connected to a three-way valve and breathing tube, and gas exchange data were acquired every 15 s. The last minute of VO_2_ and VCO_2_ data from each stage were averaged to compute RER, and these values were used to compute FOx and CHOOx in g/min using the Frayn equations [34]. Continuously, heart rate (HR) was measured using telemetry (Polar, Lake Success, NY, USA). Blood lactate concentration (BLa) was determined from a fingertip using a lancet (Owen Mumford, Inc., Marietta, GA, USA) and portable monitor (Lactate Plus, Nova Biomedical, Waltham, MA, USA) pre-exercise, during the last 10 s of cycling at 10 and 30%PPO, and 3 min post-exercise.

High intensity interval training: Participants trained three days a week on the same cycle ergometer. Initially, women warmed up for 4 min at 10%PPO. The HIIT regimen consisted of 8 (sessions 1–3), 9 (sessions 4–6), and 10 (sessions 7–9) 1 min bouts at 85%PPO with 75 s recovery between bouts at 10%PPO. This protocol is well-tolerated in sedentary individuals [17]. Heart rate (HR) was monitored throughout each session (Polar, Lake Success, NY, USA).

Monitoring of habitual food intake and physical activity: Women completed a food log for 48 h before the initial assessment of fat and CHO oxidation, in which they listed all food and drink ingested, including portion sizes, food types, and brands. This log was photocopied and returned to each participant to replicate in the 48 h before the second baseline assessment and post-test. Prior to testing on these subsequent visits, the experimenter reviewed the dietary log with each woman and asked each participant if she followed the dietary pattern used before trial 1, and all women confirmed that they did.

Participants were advised to maintain their sedentary lifestyle during the study. They were given an accelerometer (Actigraph wGT3X-BT, sampling rate = 30 Hz, epoch = 5 s, Pensacola, FL, USA) to wear on their wrist for 2 d prior to each baseline test of fat and CHO oxidation, 2 d between training sessions 4 and 5, and for 2 d prior to the final assessment of fat and CHO oxidation. They were asked to wear it 24 h per day including during sleep, with exception of when they would be in the water (swimming, the shower, etc.) when it was removed. Steps completed over each period were used to quantify habitual physical activity. Chrzanowski-Smith et al. [27] advised that objective monitoring of physical activity be performed to ensure pre-test standardization of physical activity prior to assessment of FOx and CHOOx.

Data analysis: Data are expressed as mean ± SD and were analyzed using SPSS Version 24.0 (Chicago, IL, USA). For each variable, pre-training data were expressed as the average of the two baseline tests. The Shapiro-Wilks test was used to assess normality. Independent t-test was performed to identify differences in demographic and physiological variables between groups at baseline. Repeated three-way mixed ANOVA (training = pre versus post, time = 4–6 levels assigned during progressive exercise, group = H versus C) with repeated measures was performed to identify differences in MFO, fat and CHO oxidation, HR, and BLa. The Greenhouse-Geisser correction was used to account for the sphericity assumption of unequal variances across groups. If a significant F ratio occurred, Tukey’s post hoc test was used to identify differences between means. Cohen’s d was used as an estimate of effect size (small (*d* = 0.2), medium (*d* = 0.5), and large (*d* = 0.8), and 95% confidence intervals (95%CI) were used as appropriate. Intraclass correlation coefficient (ICC) and standard error of the mean (SEM) were used to assess the reliability of baseline data. Minimum difference (MD) was calculated using this equation: SEM X 1.96 X √2 [35] to be confident, at the 95% level, that a pre to post-training change in MFO (0.08 g/min), FOx (0.08 g/min) and CHO oxidation (0.27 g/min) reflects a meaningful increase greater than the day-to-day measurement error. Pearson pairwise correlation was used to determine relationships between variables. Statistical significance was set as *p* < 0.05.

## 3. Results

No difference (*p* > 0.05) in VO_2_max, PPO, or substrate metabolism was shown between groups at baseline other than height, which was significantly higher in C versus H (Table 1). Peak HR was equal to 167 ± 10, 166 ± 11, 165 ± 11, and 165 ± 10 b/min on days 1, 3, 6, and 9 of training that overall elicited 90%HRmax. There was high compliance to training, as women completed 99.5% of all sessions. VO_2_max was increased (*p* = 0.001) by 8% in C (1.9 ± 0.4 L/min to 2.1 ± 0.3 L/min, *d* = 1.8) and 10% in H (1.7 ± 0.3 L/min to 1.9 ± 0.3 L/min, *d* = 1.5), respectively, which was similar between groups (*p* = 0.65). Body mass did not change (60.5 ± 7.4 kg vs. 60.1 ± 7.1 kg in C and 60.6 ± 8.0 kg vs. 61.2 ± 7.5 kg in H, *p* = 0.30) during the study.

Reliability data: Day-to-day variability in fat and CHO oxidation is exhibited in Table 2. The results showed no difference (*p* = 0.32–0.98) in any measure between tests as well as ICCs representing moderate to good reliability. The SEM values were relatively low and similar for FOx and CHOOx between tests. In regards to MFO, 14/22 women revealed SEM < 0.03 g/min between tests, and six of these women revealed identical MFO values. Nevertheless, an additional six women revealed differences in MFO across tests > 0.06 g/min. Coefficient of variation for MFO was equal to 30%.

Change in fat and CHO oxidation in response to HIIT: We chose to collapse our data across ethnicity as no significant interactions were revealed for maximal fat oxidation. Table 3 shows alterations in RER in response to HIIT. At baseline, RER increased during exercise (*p* < 0.001) from values at 0031%PPO equal to 0.83–0.84 to values equal to 0.96–0.97 at 50%PPO, reflecting minimal fat oxidation. Data revealed a significant training X time (*p* < 0.001) and training X time X group interaction (*p* = 0.03) but no time X group interaction (*p* = 0.86). Post hoc analyses showed that post-training RER at 10%PPO was significantly lower than at baseline in C (*d* = 0.70) and H women (*d* = 0.90). Post-training RER at 20%PPO was only lowered in H women (*d* = 0.90). At the three higher intensities, post hoc testing showed that there was no difference in post-training values compared to baseline.

Figure 2a shows the change in FOx in response to HIIT. Cycling at 10%PPO showed the highest FOx during progressive exercise after which it declined (*p* < 0.001). Time X group (*p* = 0.75) and training X group (*p* = 0.87) interactions were not shown, although there was a significant training X time (*p* = 0.03) and training X time X group interaction (*p* = 0.01). In C, post hoc analyses revealed that no post-training value of FOx was significantly greater than baseline; whereas, in H, FOx at 20%PPO (0.12 ± 0.03 g/min to 0.17 ± 0.03 g/min, *d* = 1.7) and 30%PPO (0.11 ± 0.05 g/min to 0.14 ± 0.05 g/min, *d* = 1.0) was significantly increased with training.

Results revealed a significant increase in MFO (*p* = 0.03) from baseline to post-training in C (0.19 ± 0.06 vs. 0.21 ± 0.06 g/min, *d* = 0.70) and H (0.16 ± 0.03 g/min vs. 0.19 ± 0.03 g/min, *d* = 1.3), but there was no training X group interaction (*p* = 0.92). Post-hoc analyses showed that the post-training value was significantly higher than baseline in both groups. Two C women and one H woman revealed an increase in MFO greater than 0.08 g/min, our MD value. Five women revealed meaningful increases in FOx at workloads ranging from 10–30%PPO. Nevertheless, ten additional women exhibited minimal changes in FOx or MFO of < 0.02 g/min after training (Figure 2b).

Carbohydrate oxidation increased linearly during exercise (*p* < 0.001) and peaked at 1.51 ± 0.28 g/min and 1.35 ± 0.28 g/min at 50%PPO in C and H, respectively (Figure 2c). There was a significant training X time interaction (*p* = 0.002) but training X group (*p* = 0.66), time X group (*p* = 0.46), and training X time X group interactions (*p* = 0.40) did not occur. Compared to baseline, post hoc results showed that post-training CHO oxidation was reduced at intensities ranging from 10%–40%PPO in C (*d* = 1.0032–1.5); although in H, this result only occurred at 0031%–30%PPO (*d* = 1.1–1.8). Two C and two H women showed decreases in CHOOx during exercise surpassing the MD = 0.27 g/min; whereas, the other 18 showed non-meaningful changes in CHOOx.

Change in blood lactate concentration and heart rate in response to HIIT: Blood lactate concentration increased during exercise (*p* < 0.001) and there was a main effect of training (*p* = 0.002) and training X time interaction (*p* = 0.02) (Figure 3a); however, there was no training X group (*p* = 0.30), time X group (*p* = 0.33), or training X time X group interaction (*p* = 0.63). All values differed from one another with exception of pre-exercise and that obtained at 10%PPO. Post hoc analyses showed that the post-training value at 30%PPO in both C (*d* = 0.85) and H (*d* = 0.70) was lower than pre-training.

HR increased during exercise (*p* < 0.001) and attained peak values equal to 80%HRmax. There was a main effect of training (*p* = 0.004) yet no training X time interaction was shown (*p* = 0.13) despite 5–10 b/min reductions in HR seen post versus pre-training (Figure 3b). No interactions for HR were evident (*p* = 0.22–0.47).

Habitual physical activity: There was no difference in physical activity across time (*p* = 0.26) and there was no time X group interaction (*p* = 0.56). Data are shown in Table 4. There was a significant correlation (*r* = 0.56, *p* = 0.01) between the estimate of physical activity acquired before each baseline assessment of fat and CHO oxidation, although there was no correlation (*r* = −0.36, *p* = 0.11) between the difference in physical activity and difference in MFO across these assessments.

## 4. Discussion

This study compared changes in CHO and fat oxidation in inactive C and H women performing low-volume interval training and used MD to better understand responsiveness to exercise training. Our results support our hypothesis as data exhibit similar training-induced increases in MFO between H and C women, although use of MD obtained from repeated assessments at baseline identifies a negligible amount of women exhibiting meaningful responses in this outcome. Haffner et al. [36] showed that, compared to C, H adults exhibit higher fasting and 2 h insulin concentration and lower insulin sensitivity index in response to an oral glucose tolerance test. This is concerning considering the association between impaired glucose tolerance and onset of type 2 diabetes, whose incidence is higher in H versus C [37]. As capacity for fat oxidation is related to metabolic health [23], identifying exercise-based interventions enhancing fat oxidation is important to reduce the risk of chronic disease.

Previous data in non-obese inactive women revealed a significant increase in MFO (+0.08 g/min) versus a non-exercising control group in response to nine sessions of moderate HIIT (60–70%PPO) [17]. A similar absolute increase in MFO was shown in obese men performing 8 [38] and 12 sessions of HIIT [39]. However, neither study used a non-exercising control group, fat and CHO oxidation was only assessed once at baseline, and in the Lanzi et al. study [38], physical activity and dietary intake before testing were not considered. Larger increases in FOx were reported in inactive men performing high volume MICE [15] and in inactive women completing higher volume HIIT [18], so it is possible that a larger daily dose of exercise training is needed to elicit more substantial increases in FOx. However, Nybo et al. [4] revealed no change in exercise FOx in inactive men and women completing 8 wk of HIIT (5 × 2 min bouts at 95%HRmax) despite significant increases in VO_2_max. Overall, increased FOx is not a universal response to HIIT, and this is likely due to lack of consideration of dietary intake and physical activity before testing that augment variability in the measure.

Clamp et al. [40] showed enhanced exercise fat oxidation in black South African women undergoing 12 wk of MICT, although no change in CHO oxidation was demonstrated. Similarly, our aggregate data show significant increases in MFO and fat oxidation at several intensities in response to training as well as significant reductions in CHO oxidation that corroborate prior findings [41]. However, the meaningfulness of our findings should be interpreted with caution if these changes are compared to our estimated MD. Only a small number of women (15–20%) showed true changes in fat and CHO oxidation in response to training that surpass our MD values, which consists of biological and measurement error. Higher volume interval training tends to elicit greater increases in fat oxidation [17,18], so we recommend longer-duration and higher volume protocols, especially when scientists aim to detect between-group differences in these outcomes. In addition, despite all women being inactive, VO_2_max ranged from 21–38 mL/kg/min, which in turn led to discrepant MFO values (0.08–0.25 g/min) in our sample. It is possible that a more homogenous sample would lead to lower variability and a greater ability to detect changes in fat and CHO oxidation in response to training.

Despite reliability data showing nearly identical mean values and good to strong associations between tests, there was substantial variation, especially in MFO. Similar large variability in FOx occurred in active adults completing cycling and running [25,42] and in a heterogeneous sample of adults [27]. In the present study, we used the measured values approach to identify MFO, which has similar reliability versus the polynomial model and SIN approach [27]. Our MD values stemmed from repeated baseline testing 48 to 96 h apart, similar to previous studies with the exception of Chrzanowski–Smith et al. [27] who allowed 7–28 d between tests. An alternative approach to employ would be a wash-in when participants are tested several weeks apart, perhaps using a duration similar as that required for training. However, a limitation of this approach is a greater duration for each participant in the study, which may reduce adherence. Moreover, there is greater potential for participants to change their dietary and physical activity patterns that may alter resultant data.

We attempted to better understand variability in FOx by assessing habitual physical activity before each assessment, as substantial changes in physical activity undertaken before testing may alter estimates of MFO [26]. Fletcher et al. [43] reported that physical activity is significantly associated with MFO. More recently, Amaro-Gahete et al. [44] showed in 191 sedentary adults that sedentary time-related variables were unrelated to MFO, yet there were significant associations between accelerometer-derived physical activity and MFO, although this association disappeared when adjusted for VO_2_max. Moreover, the magnitude of these associations was small, which suggests that other variables, including cardiorespiratory fitness likely have greater impact upon fat oxidation. Our data showed no differences in physical activity measured twice at baseline, as well as no association between differences in habitual physical activity recorded 48 h before each assessment and differences in MFO. Nevertheless, 8/22 women revealed marked differences (>5000 steps) in physical activity between days, yet only two of these women showed dramatic differences in MFO (0.10 and 0.12 g/min). This suggests that any variability in MFO across days may not be associated with changes in participants’ physical activity preceding testing when expressed as steps per day, and is likely due to alterations in metabolite concentrations [45], which are mediated by variations in dietary intake [43] that are not accurately reflected in a dietary log that tends to have a high number of errors [46].

Despite recent studies establishing norms for MFO in a heterogeneous population [47], no clinically meaningful increase in MFO has been identified, unlike documented changes in VO_2_max (+1 MET) [48], blood pressure (−5 mm Hg) [49], and blood glucose (−1 mM) [50], which are associated with improved health and reduced disease risk. Widespread testing of fat and CHO oxidation presents its challenges due to the dramatic variability in MFO across days and the need to set stringent dietary and physical activity requirements for participants to follow for up to 2 d pre-testing. Moreover, there is no standard graded exercise protocol, data analytic approach, or regression equation used to estimate fat and CHO oxidation, which can make comparing results across studies difficult. Nevertheless, we encourage researchers to assess MFO along with other more traditional outcomes in large-scale exercise training studies to better portray participants’ metabolic health.

Our study has some limitations. First, the results only apply to young, inactive, and non-obese women. Ethnicity was obtained through self-report, as previously used [51], and it is possible that a DNA test would be more suitable to confirm participants’ ethnicity. No measure of body fat was recorded and although women had BMI classified as healthy, it is likely that their percent body fat varied which may augment variability in our findings. Amaro-Gahete et al. [52] showed that obese adults exhibit lower fat oxidation per unit of lean mass versus normal weight adults, which exhibits the effect of body composition on the capacity for fat oxidation. We required a minimum of 6 h fast pre-testing to facilitate participant recruitment. This duration of fasting results in similar CHO oxidation, plasma insulin, and free fatty acid levels versus an 8–12 h fast in men [53]. Nevertheless, our fat oxidation values may be underestimated. We did not consider the menstrual cycle, as data show no effect of menstrual phase on MFO during graded exercise [54]. In addition, we did not recruit a non-exercising control group. However, we conducted metabolic testing twice at baseline to identify within-subject variability in our measures, and used resultant data to describe responsiveness to training, as recommended [30]. In addition, we monitored physical activity for 48 h before all assessments, and this variable did not change during the study, which gives us greater confidence in the validity and reliability of our results. However, we did not estimate the intensity of PA (i.e., moderate or vigorous) from our data, and thus we recommend that researchers elect to identify these outcomes in future work that may alter resultant determinations of fuel utilization. Similarly, we required participants to record food intake for 48 h before the initial session and replicate this pattern before subsequent sessions, although no formal analyses of these data were initiated. Our prior work [17,24] showed that macronutrient intake computed from self-reported food logs does not change (*p* > 0.05) during HIIT regimens lasting 6–12 weeks, so we are confident that it did not change in the current study. All training was supervised by experienced personnel, and HR was recorded to confirm that women did complete high intensity interval training.

## 5. Conclusions

Overall, our findings exhibit that changes in fat and CHO oxidation to low-volume HIIT are negligible, as the overall change in these outcomes was typically smaller than estimates of minimum difference. Ethnicity does not mediate the MFO response to training. To better portray metabolic adaptations to high intensity interval training, we encourage scientists to use MD obtained from repeated testing in the same population. In addition, further work is needed using long-duration and high volume exercise training to explore potential differences in fuel utilization in adults of various ethnicities due to existing health disparities as well as the strong association between fat oxidation and health status.

## Figures and Tables

**Figure 1 ijerph-18-04314-f001:**
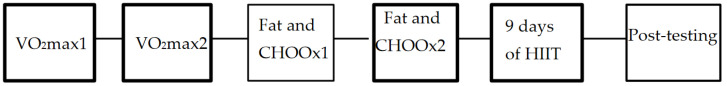
Experimental design. HIIT: high intensity interval training; CHOOx: carbohydrate oxidation.

**Figure 2 ijerph-18-04314-f002:**
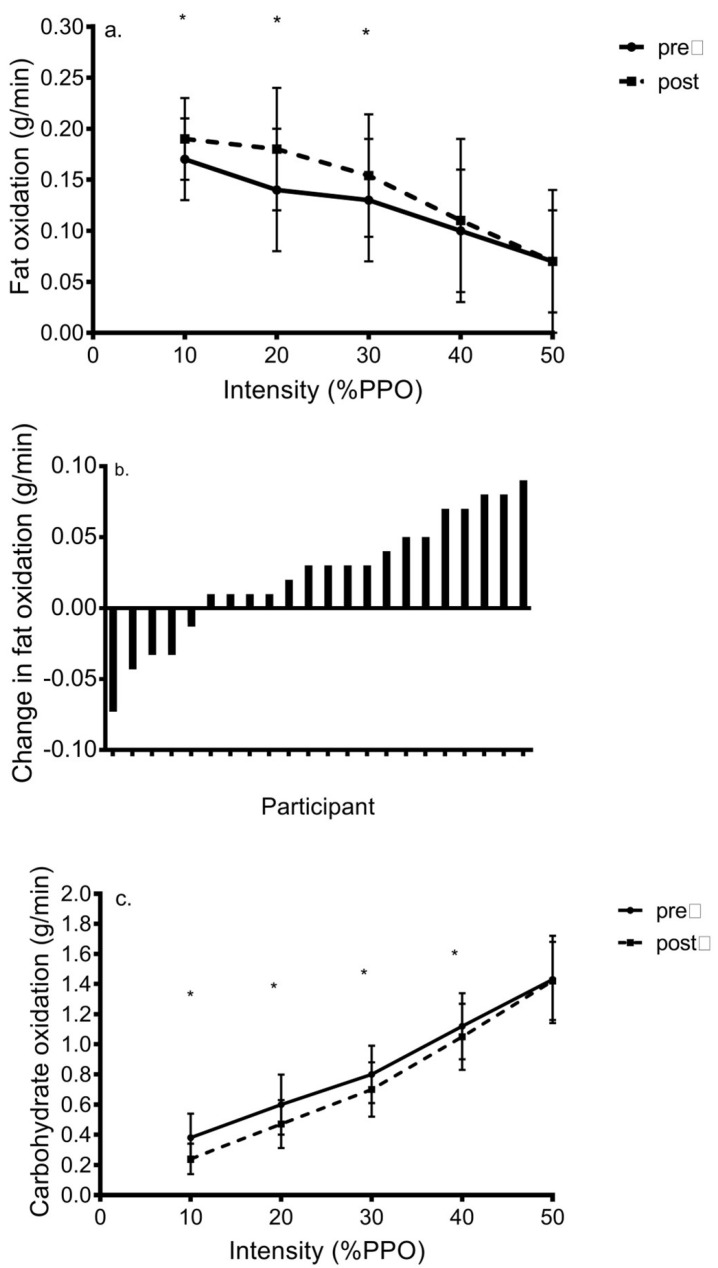
Change in (**a**) fat oxidation, (**b**) individual changes in maximal fat oxidation in response to low volume HIIT, and (**c**) CHO oxidation during progressive exercise in response to low-volume HIIT in inactive women (mean ± SD). * = *p* < 0.05 versus pre-training value.

**Figure 3 ijerph-18-04314-f003:**
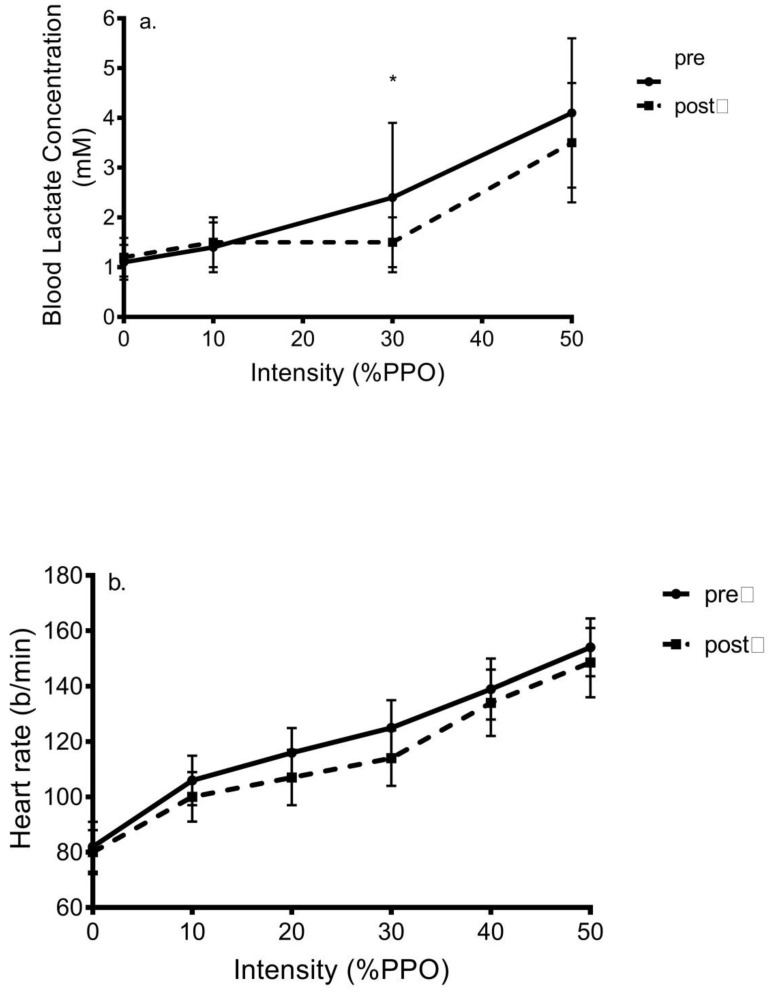
Change in (**a**) blood lactate accumulation and (**b**) heart rate during progressive exercise in response to low-volume HIIT in inactive women (mean ± SD). * = *p* < 0.05 versus pre-training value.

**Table 1 ijerph-18-04314-t001:** Participant physical characteristics (mean ± SD).

Parameter	Caucasian (n = 12)	Hispanic (n = 10)	*t*	*p*	*d*
Age (yr)	26.33 ± 5.65	22.60 ± 2.46	1.68	0.07	0.64
Height (cm)	164.79 ± 6.78	157.35 ± 5.61	3.50	0.01 *	1.28
Mass (kg)	60.49 ± 7.40	60.56 ± 8.00	0.53	0.98	0.14
BMI (kg/m^2^)	22.2 ± 2.0	24.2 ± 2.3	1.32	0.20	1.00
VO_2_max (mL/kg/min)	31.07 ± 3.74	28.37 ± 3.83	1.70	0.11	0.71
Peak Power Output (W)	175.67 ± 27.14	164.20 ± 27.95	0.97	0.34	0.43
Physical Activity (steps/2 days)	209,34.91 ± 3703.25	18,562.67 ± 7491.68	0.92	0.37	0.43

BMI = body mass index; VO_2_max = maximal oxygen uptake; * = *p* < 0.05 between groups.

**Table 2 ijerph-18-04314-t002:** Average MFO (maximal fat oxidation), fat and carbohydrate oxidation, intraclass correlation coefficient, standard error of the mean, and limits of agreement from repeated progressive exercise tests performed at baseline by inactive women at intensities ranging from 10%–50%PPO (mean ± SD). ICC: Intraclass correlation coefficient; SEM: standard error of the mean; LOA = limits of agreement; EE = energy expenditure.

Variable	Test 1	Test 2	*p* Value	ICC	SEM	LOA
MFO (g/min)	0.19 ± 0.06	0.19 ± 0.06	0.61	0.81	0.03	0.07–0.31
FOx_10%_ (g/min)	0.18 ± 0.06	0.18 ± 0.06	0.98	0.79	0.03	0.06–0.30
FOx_20%_ (g/min)	0.14 ± 0.07	0.15 ± 0.08	0.73	0.72	0.04	0.01–0.28
FOx_30%_ (g/min)	0.13 ± 0.06	0.14 ± 0.08	0.76	0.76	0.03	0.00–0.27
FOx_40%_ (g/min)	0.10 ± 0.07	0.11 ± 0.07	0.32	0.84	0.03	−0.03–0.24
FOx_50%_ (g/min)	0.08 ± 0.07	0.08 ± 0.06	0.85	0.72	0.03	−0.04–0.20
CHOOx_10%_ (g/min)	0.36 ± 0.21	0.38 ± 0.24	0.69	0.79	0.10	−0.06–0.80
CHOOx_20%_ (g/min)	0.59 ± 0.21	0.63 ± 0.24	0.52	0.81	0.10	0.18–1.03
CHOOx_30%_ (g/min)	0.80 ± 0.20	0.78 ± 0.27	0.70	0.85	0.09	0.34–1.24
CHOOx_40%_ (g/min)	1.11 ± 0.23	1.10 ± 0.24	0.67	0.85	0.09	0.75–1.55
CHOOx_50%_ (g/min)	1.42 ± 0.30	1.41 ± 0.33	0.77	0.88	0.11	0.80–2.02
EE (kcal)	107.4 ± 14.8	108.5 ± 17.1	0.47	0.95	3.5	76.6–139.4

**Table 3 ijerph-18-04314-t003:** Change in respiratory exchange ratio in response to low-volume HIIT (high intensity interval training) in inactive women (mean ± SD).

Workload	Pre	Post
10%PPO	0.84 ± 0.04	0.80 ± 0.03 *
20%PPO	0.88 ± 0.05	0.85 ± 0.04 *
30%PPO	0.91 ± 0.04	0.89 ± 0.04
40%PPO	0.95 ± 0.04	0.94 ± 0.05
50%PPO	0.97 ± 0.04	0.97 ± 0.04

* = *p* < 0.05 versus pre-training value.

**Table 4 ijerph-18-04314-t004:** Habitual physical activity recorded over a 48 h period in inactive women performing low-volume HIIT (mean ± SD).

Variable	Baseline 1	Baseline 2	Mid	Post-Training
Steps	21,346 ± 5154	18,950 ± 5905	20,964 ± 5615	17,235 ± 4963

## Data Availability

All relevant data pertinent to this study is presented in the Results section.

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
