# Peer review of "Effect of Ethnicity on Changes in Fat and Carbohydrate Oxidation in Response to Short-Term High Intensity Interval Training (HIIT): A Pilot Study"

_ijerph, 2021, doi:10.3390/ijerph18084314_

Round 1
Reviewer 1 Report
This is an interesting study assessing the effect of HIIT in changes in fat and carbohydrate oxidation. The study novelty might be related to ethnicity. However, it is not clear in the research gap. The authors may include a paragraph for the research gap. The main limitation of this study is the absence of a control group. That may lead to type I or II errors with misleading interpretations. The authors may well state that this results are only representative of this sample.
Abstract: Abstracts with many abbreviations are hard to read and understand.
L.13: What is FOx? Fat oxidation?
L.18: What is MFO? Maximal Fat Oxidation?
L31: This reviewer did not understood if the authors are referring to the population levels or if the guidelines are low.
L.45 Typo Error.
L.50: This reviewer has some doubts about this variability. How was it measured? This can lead to misleading understanding. This statement almost justify that 50% of differences between participants are due genetics... Please review the sentence and better explain this 50%.
L.55-57: Where's the reference?
L.83: Typo error (
L.88: A new paragraph should be included with an "altogether" of the above mentioned information. Then the research gap must be included. No studies with this populations (C and H)?
L.102: Please include a flowchart in study design.
L.112: The main limitation of this study is the absence of a control group. This study should have control groups for H and C.
L.128: Part of table 1 is results... The authors should include effect sizes.
L.204: The authors may include the effect sizes.
L.214: is the -0.95 the r of Pearsons test?
L.321: Before paragraph 2, a new paragraph must be included. The new paragraph may justify the chosen tests. Why those tests and what's their validity in comparison to more precise and accurate methodologies?
L.396: What's the probability of a ancestor influence the ethnicity?
L.398: The absence of control groups is the main limitation.
Author Response
Please see attachment--thank you.

Reviewer 2 Report
Thank you for your submission to IJERPH. The manuscript reports on the effects of ethnicity on changes in fat and carbohydrate oxidation
in response to short-term high intensity interval training
(HIIT). Given the small sample size the authors note that this is a pilot study. The results showed that low-volume HIIT does not alter fat and CHO oxidation and there is little effect of ethnicity on the response to training. This manuscript is clear and concise, and the authors show a revealing command of future studies. The authors have done a great job of providing an informative and meaningful addition to the current study field.
However, there are some changes that the authors are encouraged to revise to elevate the overall contribution of the paper to this research field.
Abstract - line 13 please define FOx.
Intro - very clear and concise, forms a straightforward research question.
Line 96 - should this read C at the end of the sentence?
Is there a specific reason for only examining women in this study? Consider adding some info regarding this to the intro.
Methods - line 107 - which survey was used?
Line 107-108 - performed used twice in this sentence.
Line 116 - p in italics? (similar throughout paper)
Would it be beneficial to add mean and SD BMI to table 1? Why was %BF data not collected?
Do you have HR data from the HIIT sessions to ensure participants were reaching the correct intensity?
Was the Hans Rudolph mask also used during the VO2 max assessment?
Line 168 - missing reference
Line 213 - consider adding body mass information for the reader
Table 4 - consider adding "steps" or "step count" somewhere to this table so it reaffirms this to the reader. It is also interesting that these inactive women are getting approximately 10,000 steps/day across all timepoints?
Intro vs Discussion - you use H and C throughout the entire introduction but this changes in the discussion - consider being consistent with one or the other
References 1-5 and 50-52 need consistency in formatting
Author Response
Please see attachment--thank you.

Round 2
Reviewer 1 Report
As mentioned before, this is an interesting study assessing the effect of HIIT in changes in fat and carbohydrate oxidation. The authors have improved the manuscript. However the results presentation is poor. As reviewer, I usually consider typo errors along the manuscript due to lack of attention. However, in some cases it is more than sufficient to reject a paper. Please be careful in future research.
L.46: Typo error: Gurd et al. (10]. The parenthesis.
L115: Below the figure.
Results: The authors may present the T values. It is not acceptable an article in IJERPH only with p values.
136: Please include effect sizes. However, the comparisons should be in results section.
L.217: This is a significant difference. There are usually differences, but this one has significance. Review the full article for these cases.
Table 3: Table incomplete. Include test value, p and effect sizes.
Discussion should start with the aim of the study, the hypothesis and if the results supports or reject the hypothesis.
Second paragraph should be about the methods choice. The reason why for each method.
Please recommend further research.
Author Response
Please see attached document containing a revised point-by-point rebuttal to new concerns raised by Reviewer #1; thank you.
